# Compound climate events transform electrical power shortfall risk in the Pacific Northwest

S.W.D. Turner [1], N. Voisin [1,2], J. Fazio[3], D. Hua[3] & M. Jourabchi[3]

Power system reliability is sensitive to climate-driven variations in both energy demand and water availability, yet the combined effect of these impacts is rarely evaluated. Here we show that combined climate change impacts on loads and hydropower generation may have a transformative effect on the nature and seasonality of power shortfall risk in the U.S. Pacific Northwest. Under climate change, potential shortfall events occur more readily, but are significantly less severe in nature. A seasonal reversal in shortfall risk occurs: winter shortfalls are eradicated due to reduced building heating demands, while summer shortfalls multiply as increased peak loads for day-time cooling coincide with impaired hydropower generation. Many of these summer shortfalls go unregistered when climate change impacts on loads and hydropower dispatch are analyzed in isolation—highlighting an important role of compound events.

[1] Pacific Northwest National Laboratory, Seattle Research Center, 1100 Dexter Ave N., Suite 500, Seattle, WA 98109, USA. [2] Civil and Environmental Engineering, University of Washington, 201 More Hall, Box 352700, Seattle, WA 98195-2700, USA. [3] Northwest Power and Conservation Council, 851 S.W. Sixth Avenue, Suite 1100, Portland, OR 97204, USA. Correspondence and requests for materials should be addressed to N.V. (email: nathalie.voisin@pnnl.gov)

A power system comprises a fleet of electricity generators and a transmission system that links those generators to electricity users—including homes, businesses, and industry. The system is planned and operated to avoid shortfalls between electricity demand and supply, and it is deemed adequate if shortfall risk is managed at a sufficiently low level. Planners typically schedule new generating capacity investments to deal with the effects of projected socioeconomic change (e.g., population growth) and capacity retirements, although there is a growing body of science literature highlighting the mechanisms by which climate change may affect power shortfall risk. On the supply side, changes to streamflow could affect hydropower generation[1–3] and thermal plant cooling, leading to capacity derating at individual plants[4–6]. On the demand side, warming temperatures are likely to affect power loads for building heating and cooling[7–10]. Notwithstanding the important insights delivered by studying these individual phenomena, a deeper form of analysis is needed if results are to inform practical power system policy and planning. First, climate risk may be misrepresented if the assessment fails to incorporate the dynamics of the whole, interconnected power system[11–13]. For example, climate change may impair the generating capability of a particular resource type, but the associated impact may be absorbed if it occurs during a non-peak demand season, or if resulting generating loss can be satisfied from alternative resources or purchased from adjacent networks. Climate impacts must therefore be placed in a broader systems context by studying the response of regional power supply networks. A second complexity—rarely assessed—is the potential for compound events caused by multiple interacting climate impacts. For example, a trend of warmer, drier summers with increased occurrence of heatwave and drought conditions may cause higher peak loads and reduced water availability simultaneously. Studied separately, these impacts may be insufficient to register concern. But taken in combination, they may cause severe power shortfalls. The possible threat of multiple climate-related phenomena acting in combination is a well-versed hypothesis brought forward by climate scientists[14,15] but not yet tested using a power system model.

Here we investigate the combined supply and demand impacts of climate change on the adequacy of the U.S. Pacific Northwest power system. This region depends heavily on hydroelectric power from more than 130 dams in Columbia River Basin, which collectively provide about half of overall annual generating capability[16]. Water availability is snow-melt driven with augmented flows lasting into late spring, causing a strong seasonal signal in hydropower dispatch potential. A diverse mixture of natural gas, coal, and renewables (predominantly recently-installed wind) contribute most of the remaining supply capability (Fig. 1a, b). Peak loads tend to occur during region-wide, winter cold-snaps (building heating) and to a lesser extent during hot summer days (building cooling). The phenomena of interest in this region are therefore temperature effects on winter heating and summer cooling demands, and the temperature and precipitation effects on the snow-melt driven hydrological regime that controls seasonal hydropower availability[17].

We evaluate the ability of the Pacific Northwest power system to reliably generate and distribute electricity under two potential infrastructure portfolios for the year 2035 (Fig. 1c), which is a relevant horizon for current planning decisions. We perform this assessment using GENESYS[18]—an hourly resolution, economic dispatch model. GENESYS simulates detailed constrained dispatch of regulated hydroelectric power projects in the Columbia River Basin and of regional thermal plants alongside an extra-regional import market. We run six separate studies derived from two infrastructure expansion cases and three climate scenarios—one that neglects climate change and two that follow conservative

climate change projections for the 2030s. The climate change scenarios are based on bias-corrected, downscaled climate projections from global atmospheric models INMCM-4 and GDFL-ESM2M, which are chosen because they represent lower-bound (conservative) estimates of climate change in the Pacific Northwest[19]. The global atmospheric models are simulated with RCP 8.5 radiative forcing[20]. (Note that RCP choice has negligible influence on warming in the Pacific Northwest region within the relatively short time horizon considered in this work; see Supplementary Figures 1, 2). Hourly power demands are projected using an econometric temperature-load model used by the Northwest Power and Conservation Council, while water availability profiles are computed using the Variable Infiltration Capacity model[21] (see Methods).

Monte-Carlo simulation with 6000 resamples is used in each GENESYS study to reach realistic extreme conditions under which shortfall may occur, arising from combined effects of load and streamflow as well as wind and forced outages of thermal generation plants (for which possible climate impacts are excluded). Resulting GENESYS-simulated records of potential power supply shortfalls are used to visualize performance and to compute a suite of adequacy metrics, following the conception that a well performing system fails marginally and seldom, whilst a poorly performing system fails substantially and often. These metrics include the Loss of Load Probability (the proportion of simulated years that contain at least one shortfall event), Average Event Duration (the mean duration of all simulated shortfall events, in hours), and Average Maximum Shortfall (the mean maximum hourly shortfall across all shortfall events, in MW) (see Methods—Performance Measures).

Our results show a transformative effect of climate change on power shortfall risk by the 2030s, with the frequency, severity, and seasonal distribution of shortfall events affected significantly. Shortfalls occur more readily under climate change, while average event magnitude and duration are markedly reduced. A dramatic seasonal switch in the concentration of shortfall events arises as the impacts of increased power demands and impaired hydropower availability coincide in summer months.

## Results

**Risks and opportunities from climate change**. We find that modest climate change has a profound impact on the performance of the U.S. Pacific Northwest power system in the year 2035. Assuming existing infrastructure expansion policy, the loss of load probability is at least doubled—meaning twice the probability of incurring shortfall in any year (Fig. 2). By current regional planning standards—which recommend a target annual loss of load probability of 5%—these impacts warrant significant new investments to expand capacity. Yet to focus on probability of failure alone is too narrow a criterion for power system planning[22]; it is equally important to incorporate the potential damage caused by each event, which is a function of shortfall duration and magnitude. We find that whilst shortfall events occur more frequently under climate change, the nature of those events is more amenable. The average event lasts about half as long (~13±1 to ~7±1 h) and is significantly less intense (average maximum shortfall cut from ~1000 to ~400 MW). Climate change may therefore be viewed as both a risk and an opportunity for power system performance, depending on one's estimation of damage and ability to adjust operations in relation to shortfall duration and magnitude.

**A seasonal shift in power shortfall risk**. The impact of climate change on annually measured performance metrics can be further understood by investigating seasonal differences. Neglecting

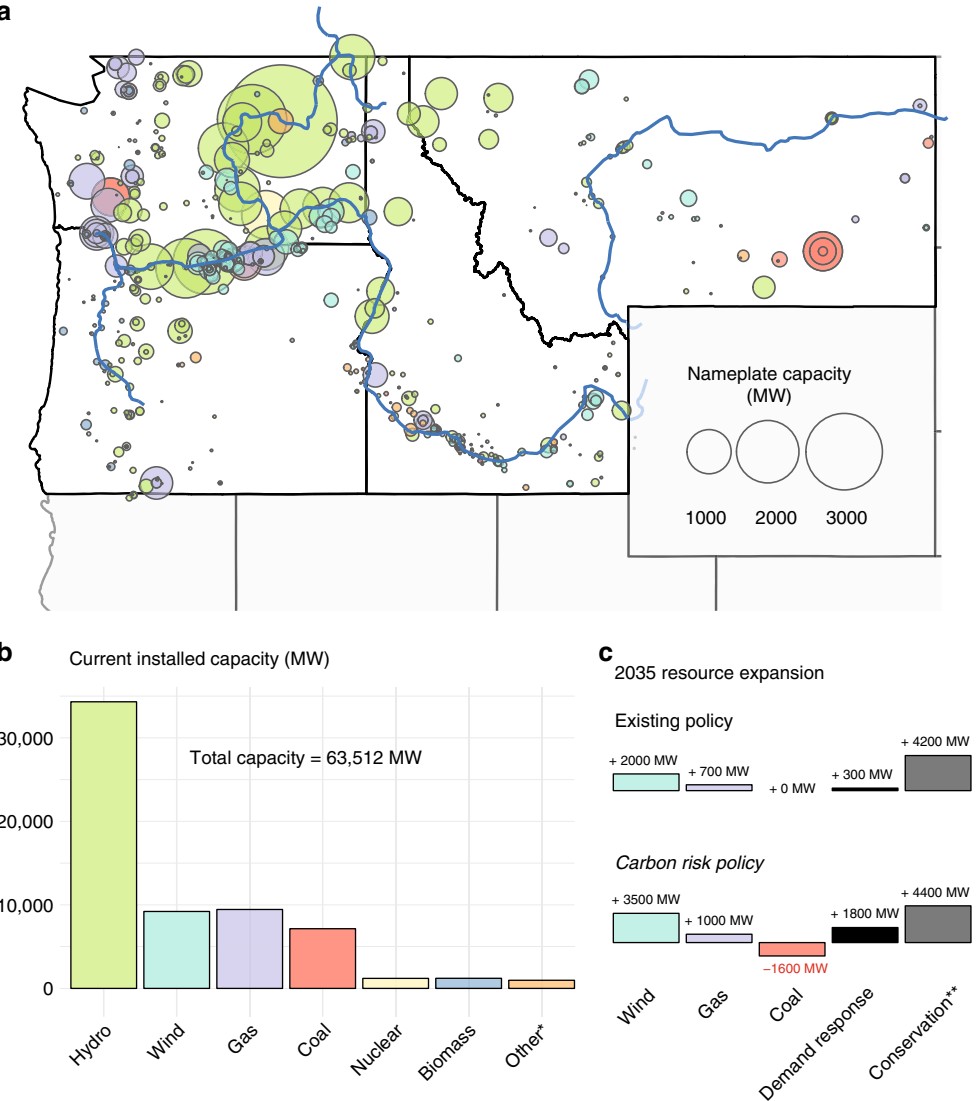

**Fig. 1** Power plants by resource type in the U.S. Pacific Northwest. *Other comprises petroleum, solar, geothermal, and energy storage (pumped storage hydropower and battery). **Conservation is measured in MWa (i.e., annual average generation), not MW (nameplate capacity)

climate change, we find that winter events dominate the simulated shortage record, occurring three to four times more often than summer events—and with significantly greater severity and duration (Fig. 3). This reflects the generally higher wintertime loads in the U.S. Pacific Northwest, where heating systems are ubiquitous and air conditioning systems nonessential (summer conditions of the major load centers of the region are mild, so most homes lack central air conditioning and will generally use window units only during severe heatwaves). Perhaps surprisingly, the modest warming imposed by our chosen climate change scenarios causes a dramatic seasonal shift in vulnerability. Winter power shortfall events are almost obliterated in the climate change simulations, primarily due to a reduction in peak loads with warmer temperatures (water availability is actually increased marginally under the INMCM climate projection, resulting in removal of all winter events in the simulations). Meanwhile, short-duration summer events multiply. The chance of incurring power shortfall in September of any year jumps from 0.3% to 3–4% in the existing policy case, for example. As the reversal in seasonal vulnerability takes place, the long and severe events associated with winter failure are replaced by short, low-magnitude events associated with summer—causing the

significant changes in annual average event duration and magnitude reported above. This seasonal disparity in event type and associated climate impact results in an apparently unstable situation in which modest warming flips both the nature and timing of shortfall vulnerability.

**Influence of compound climate events**. The influence of compound events is evident in these results (i.e., events that materialize only when water availability and load change impacts coincide). These events occur primarily during summer months, arising when increased peak load coincides with impaired hydropower (for the modeled scenarios, water availability has a very marginal effect on winter hydropower potential). Figure 4 shows that the sum of the additional summer shortfalls caused by load and water availability impacts in isolation is about half the number of additional shortfalls incurred when these impacts are combined. So we may say that about half of the overall summer climate risk increase is due to compound events.

Summer compound events are evident in both the existing and carbon risk policy cases, although the carbon risk policy incurs far fewer events overall (Fig. 4b). This is a result of more favorable

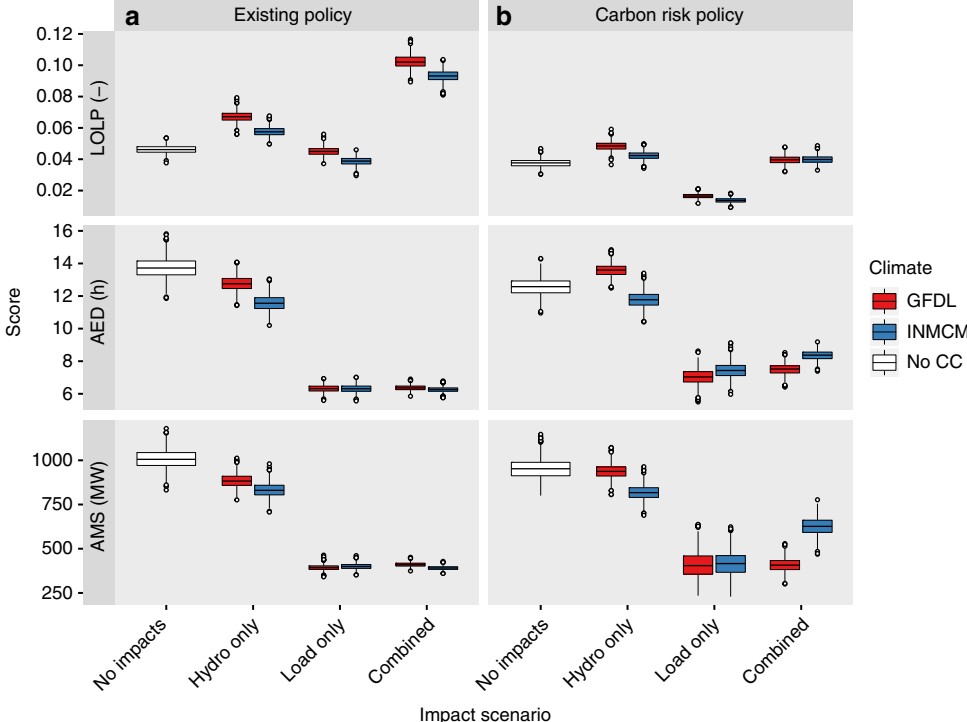

**Fig. 2** Power system adequacy with 2035 infrastructure. Results are given for existing and carbon risk policy scenarios. Metrics are loss of load probability (LOLP), average event duration (AED), and average maximum shortfall (AMS). Uncertainty distributions are derived for each case using a bootstrap with 1000 repetitions. Boxplots follow the Tukey convention, with five summary statistics (median, 25th and 75th percentiles, 1.5* interquartile range) and all outlier points

demand response incorporated into the carbon risk system set up. With less investment in fossil resources, the carbon risk scenario requires significant flexibility in the system to meet required resource adequacy in 2035. This flexibility is achieved through demand response measures that curtail power supply to non-critical demands during peak events (achieved through agreements between utilities and customers to reduce demand for electricity during periods of stress—see Methods). The result is that the system designed for adequate power supply in the no climate change scenario is more vulnerable in winter than summer. Since winter shortfall risk is alleviated by climate change, the carbon risk set up—which is better adapted to summer shortfall risk—ends up being more resilient overall. So a carbon risk policy adapts better not only to the economic risks of a carbon tax, but also the modest climate change impacts on summer load and hydro dispatch explored in this work.

## Discussion

A planner or policy-maker trying to interpret and respond to these results might consider the following practical implications. First, the dramatic shift in vulnerability of the system from winter to summer highlights the importance of informing resource expansion planning with the combined effect of climate change on load and water availability. For instance, there are certain actions that might be taken in light of the potential for increased summer event frequency. Demand response measures may be viewed more favorably, and alternative joint power-river operating methods could be explored. Second, given the apparent sensitivity of shortfall severity to climate change, it may be useful to know how damage or cost varies according to shortfall duration and magnitude. This has to be studied in light of the potential for non-linearity in costs of emergency actions (e.g., compensation to industry for loss of power). For example, a 5000 MW shortfall is likely to be more than five times as costly as a

1000 MW shortfall if it exhausts standby and emergency options, causing rolling brown-outs for customers. Incorporating this level of understanding will be necessary to make risk-based planning decisions based on an analysis of shortfall events. Third, it will be important to consider uncertainty in the load and hydropower dispatch projections, and the conditions under which the forcing effects used in this study could be mitigated or exacerbated. Important considerations might include evolving market and regulatory structures that promote inter-regional planning, transmission, and distribution constraints, changes to the Columbia River Treaty, and new environmental regulations. There may also be socioeconomic changes that affect the relationship between load and temperature. Homes that currently lack electrified cooling may install new air conditioning systems. The prevalence of air conditioning systems in Portland (Oregon,), for example, increased from ~44% of households in 2002 to more than 70% of households in 2016[23]. The future relationship between temperature and load will depend on whether and for how long this uptake trend will continue before reaching saturation.

The Western electricity crisis of 2001 was referred to as a "perfect storm"—the culmination of a decade of underinvestment and a steadily widening supply-demand deficit, exposed by an extreme dry period with impaired hydropower conditions in the Pacific Northwest[24]. The crisis wrought enormous financial costs for utilities in the Pacific Northwest, including substantial compensation paid to the aluminum industry to halt production. Today, rigorous, forward-looking planning ensures that power systems are built out to be more resilient for now and in the future. Yet whilst planning makes allowances for expected socioeconomic change, including population growth and industrial development, the possible impacts of climate change on shortfall risk are rarely evaluated. Currently, the science literature contains only a small and piecemeal collection of studies

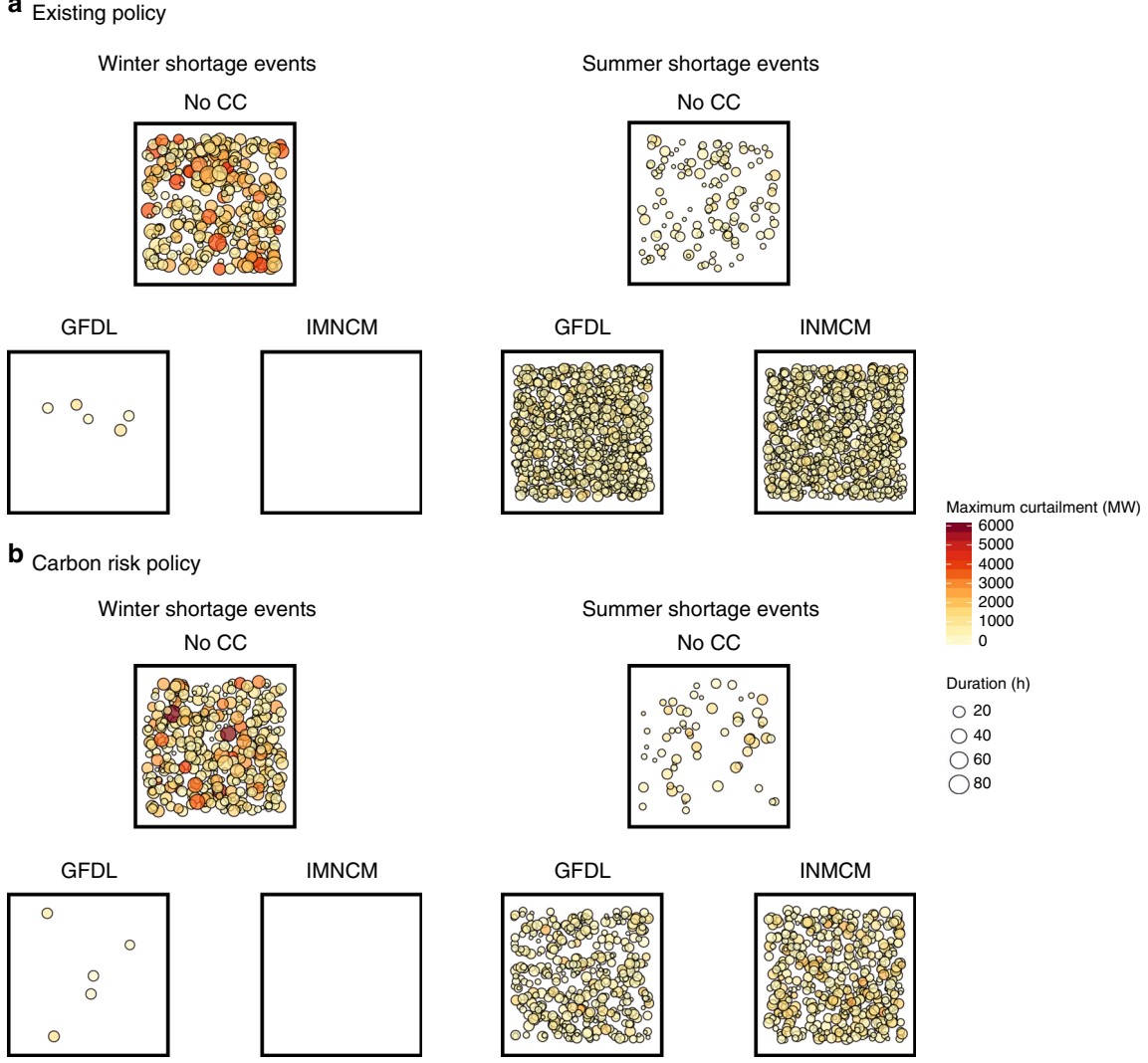

**Fig. 3** Power shortfall events in 2035. Each point represents a simulated shortfall event within either winter or summer from 6000 one-year simulations. Number of points indicates frequency of occurrence, whilst size and color give duration and maximum curtailment, respectively. Points are positioned randomly inside each box

examining climate impacts on grid operations, with each study focusing on disparate aspects of the power system. For example, a study of the U.S. Eastern Interconnection found that grid reliability is sensitive to summer heatwaves, which may raise peak cooling loads and derate gas turbine plants simultaneously[25]. Similarly, a study of grid operators in Germany and Austria demonstrated a need for additional generating capacity to meet growing summer peak loads for cooling[26]. The importance of compound effects demonstrated in the present work suggests that what planners will ultimately need is an assessment that combines all potential climate-related impacts across a spectrum of climate futures and policy scenarios.

In this study we apply lower-bound, conservative climate change scenarios to demonstrate a measurable impact of compound events under the most conservative climate scenarios available. Summer shortfall risk would be intensified further under more extreme forcing that is projected by other models, or which may be expected later in the century (see Supplementary Figure 1). Planning for such effects will be particularly important for the Pacific Northwest, which, like many other power grids throughout the world, exports electricity across adjacent grids. Further research for this region may be directed toward improved

understanding of climate impacts on supply and demand under a broader range of climate futures, including the imminent river flows and reservoir rule curves to be produced under the auspices of the River Management Joint Operating Committee (RMJOC).

## Methods
**Economic dispatch model**. The Generation Evaluation System (GENESYS) model, developed by the Northwest Power and Conservation Council, is a Monte-Carlo computer program that performs a chronological hourly simulation of the Pacific Northwest power supply for a single operating year (October through September, 8760 h)[18]. For a typical resource adequacy study, thousands of simulations are run (6000 simulations in the present work), with each simulation drawing a different combination of four random variables: temperature-sensitive loads, temperature-correlated wind generation, generator forced shortfalls, and unregulated modified river flows, each of which is briefly described below.

Temperature-sensitive hourly loads for a specific future year are produced by the Council's econometric load forecasting model (see following section), which uses historical data to project future load growth and energy efficiency savings. The model creates 88 sets of 8760 temperature-sensitive hourly loads based on 88 years of historical daily average temperatures at the four major load centers in the region (Seattle, Portland, Spokane and Boise). At the beginning of each simulation, all of the year's hourly loads are fixed by drawing from one of the 88 possibilities. Hourly loads are then adjusted for firm out-of-region contracts (i.e., exported energy is added to the load and imported energy is subtracted) and wind generation, which is modeled as a load-reduction resource and is subtracted from the load.

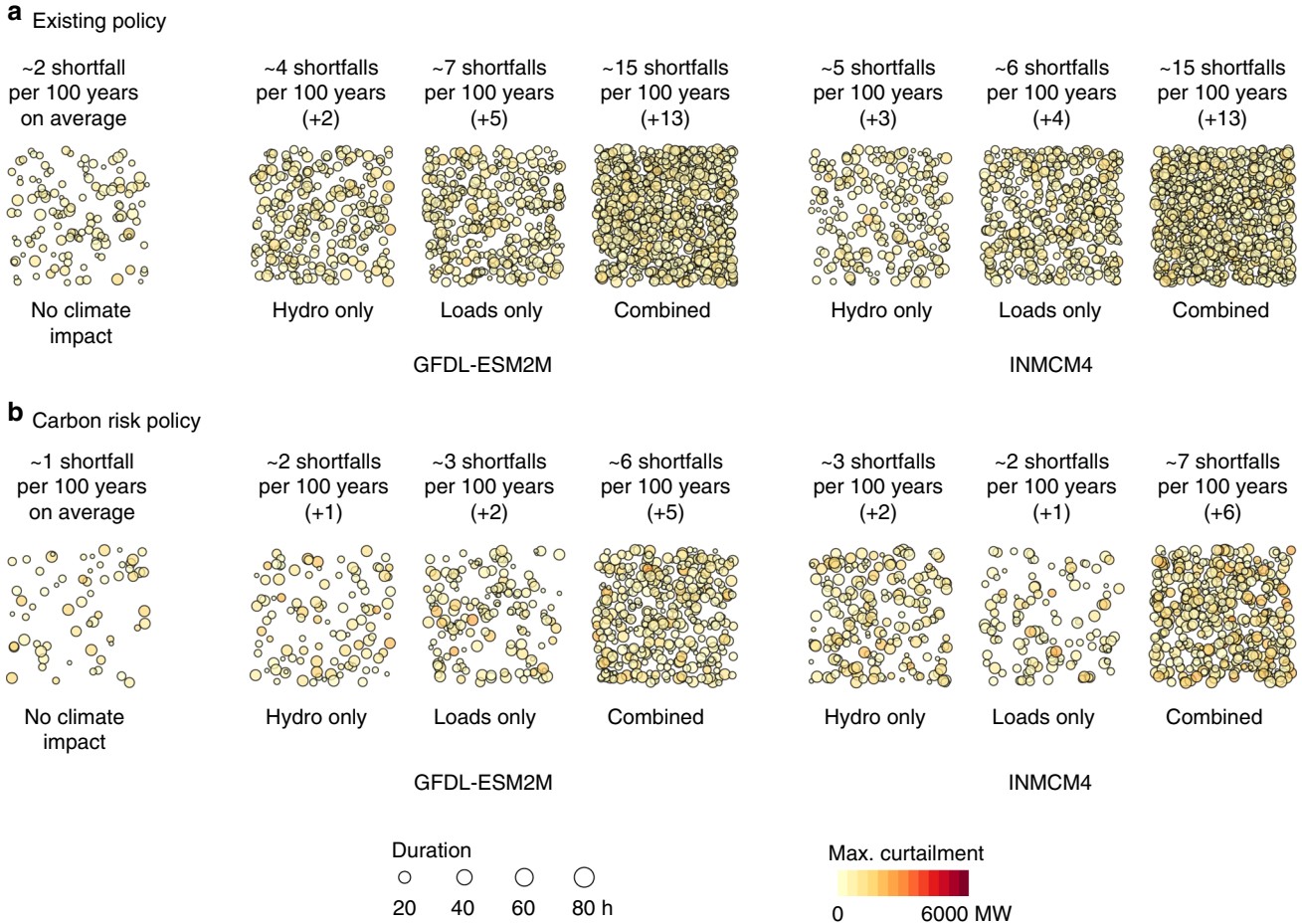

**Fig. 4** Power shortfall events with isolated and combined impacts. Results are given for summer 2035 for existing policy and carbon risk policy cases. Each point represents a simulated shortfall event out of 6000 simulations. Number of points indicates frequency of occurrence, whilst size and color give duration and maximum curtailment, respectively. Points are positioned randomly inside each box

Most of the wind generation in the Northwest (located in the Columbia River Gorge) has been shown to be weakly correlated to temperature. More precisely, as temperatures get extreme (very hot or very cold), wind generation tends to be low. Using historical temperatures and observed wind generation, bootstrap statistical methods are used to create synthetic hourly capacity factors for all 88 temperature-year profiles. Furthermore, to better capture the effects of wind generation uncertainty, 20 possible sets of hourly capacity factors are created for each temperature-year profile. Thus, at the beginning of each simulation, after the temperature-year profile is chosen, all 8760 hourly wind capacity factors are fixed by drawing from one of the 20 possible sets for that temperature year. Hourly wind generation is calculated by multiplying the hourly capacity factor by the amount of installed wind nameplate capacity and, as mentioned above, then subtracted from the hourly loads.

Forced outages are modeled separately for thermal and hydroelectric generation. Thermal generator forced outages are accounted for dynamically during the simulation, in contrast to load and wind generation whose hourly values for the entire year are selected at the beginning of each simulation. GENESYS simulates the hourly operation of individual thermal generators based on their operating characteristics and constraints. The availability of each thermal resource is determined from its forced-outage, mean-time-to-repair and mean-time-between-failure rates[27]. Hydroelectric generator forced outages and maintenance are accounted for by applying fixed monthly availability factors.

Finally, unregulated inflows at all hydroelectric projects are drawn from a set of 80 historical water-year profiles. Just as the hourly loads and wind generation for the entire year are set at the beginning of each simulation by the temperature-year selection, unregulated monthly inflows at each hydroelectric project are set to the historical water-year profile selection. For each project, the selected inflow data also include water withdrawals as well as operating guidelines such as maximum (flood control) and minimum end-of-month elevation limits, ramping rates, maximum and minimum outflow limits, and other operating constraints. From these data, monthly generation at each project is calculated and then summed to get the total hydropower system generation. At the beginning of each month, the GENESYS model estimates the amount of hydroelectric energy to be dispatched based on its assigned operating cost relative to the operating costs of other resources. Monthly hydroelectric energy is then allocated across the hours of each day based on load shape and is limited by operating constraints and maximum sustained-peaking capability. Additional hourly hydroelectric generation can be dispatched, above the allocated amount, if all other resources are fully dispatched and a shortfall still exists. At the end of each month, the final amount of hydroelectric energy dispatched is used to adjust initial reservoir elevations for the next month.

**Policy case description**. Resource expansion policies are based on two plausible, diverging carbon policy futures. For each case, a possible suite of resource types is developed to match the relevant policy story. The existing policy case represents a future for which the existing resource planning policies and standards are continued. This means that policy incorporates current federal and state policies such as renewable portfolio standards, new plant emissions standards, and renewable energy credits. This policy disregards possible future additional carbon dioxide regulatory costs and economic risks, including actions that U.S. Northwest states could take in order to comply with recently finalized limits on carbon dioxide emissions from existing power generation. The carbon risk policy represents a resource expansion trajectory that addresses the economic risks implied by carbon dioxide reduction policies additional to those incorporated in the existing policy case. Specific resource acquisitions given in Fig. 1c originate from the Northwest Power and Conservation Council's Seventh Power Plan[16]. The economic risk of a carbon tax imposes on the carbon risk policy a need to reduce power from fossil resources. The hallmark of this policy is therefore a retirement of coal and coinciding expansion of low carbon and carbon-neutral measures. This includes additional wind and significantly expanded demand response, which includes load curtailment agreements and standby resources. Both policies make a significant allowance for conservation measures.

**Temperature effects on load**. Climate change impacts on load are derived using an econometric, temperature-load model that is deployed routinely by the Northwest Power and Conservation Council[28]. The model relates loads to a set of parameters that include season, day of week, holidays, employment, population, conservation targets, and temperature deviations from normal. The model is

calibrated to return the load for a given day of the calendar year subject to deviation in the day's temperature relative to a historical mean for that particular day (measured across 1928–2016, using a population-weighted, average temperature for the Pacific Northwest region). To illustrate: if we wanted the load for a given temperature on January 1st, we would feed the model with the difference between that temperature and the mean temperature measured across all occurrences of January 1st from 1928 to 2016. To incorporate climate change impacts on temperature, we adopt the following procedure. First, average daily temperatures are computed from GCM projections for the period 2020–2049—meaning we compute the average temperature, across all thirty years, for Jan 1st, Jan 2nd, … Dec 31st. These daily averages are then transformed to a series of 365 deviations from the historical mean daily temperature for each calendar day. We then add these deviations to the full historical series of daily temperature deviations (1928–2016), resulting in an 88-year daily time series of temperature deviations representing a 2030s climate, but with variability closely resembling history. The updated set of daily temperature deviations are fed into the econometric load model, giving an 88-year time series of daily loads under climate change. Finally, these loads are disaggregated to hourly resolution using historical hourly load shapes. Ahead of assessing climate impacts on loads, year 2035 conservation targets and projected employment and population estimates are updated in the model.

**Climate impacts on hydropower.** Flow data used in the GENESYS adequacy model are modified—meaning they include the effects of irrigation and evaporation. However, the available climate-change flows[29] do not include effects of irrigation or evaporation. We deal with this discrepancy the following way. Let the set of observed historical modified flows currently used in GENESYS be $\{q_{mj}^M\}$ where subscript $j \in (1929, \dots, 2008)$ is the historical water-year index, and subscript $m \in (1, \dots, 14)$ is the monthly period index: specifically, $m = 1$ represents October and $m = 14$ represents the following September, with April and August each having two (half-month) period indices due to flows generally having the largest intra-monthly variations for those months. Similarly $\{q_{mj}^N\}$ is the corresponding set of observed historical non-modified flows. Let the set of climate-change non-modified flows and modified flows be respectively $\{Q_{mj}^N\}$ (available) and $\{Q_{mj}^M\}$ (not yet available), where $j \in (1950, \dots, 2099)$ is the climate-change water-year index subscript, and subscript $m$ is the previously described monthly-period index. An approximation for $\{Q_{mj}^M\}$ could be obtained as follows.

Let $\overline{Q_m^{NH}}$ and $\overline{Q_m^{NF}}$ be respectively the averages of non-modified flows $\{Q_{mj}^N\}$ over historical water-years (1950–2016), and 2030s water-years (2019–2049). Similarly, let $\overline{Q_m^{MH}}$ and $\overline{Q_m^{MF}}$ be respectively the averages of modified flows $\{Q_{mj}^M\}$ over historical water-years (1950–2016), and 2030s water-years (2019–2049). We first assume that effects of irrigation and evaporation (i.e., differences between non-modified and modified flows) on average are about the same for both historical and 2030s water-years. Therefore, $(\overline{Q_m^{NF}} - \overline{Q_m^{MF}}) \approx (\overline{Q_m^{NH}} - \overline{Q_m^{MH}})$ which leads to $\overline{Q_m^{MF}} \approx \overline{Q_m^{MH}} + (\overline{Q_m^{NF}} - \overline{Q_m^{NH}})$. We further assume that $\overline{Q_m^{MH}}$, the climate-change modified flows averaged over water-years 1950–2016, are approximately equal to $\overline{q_m^M}$, the observed modified flows averaged over water-years 1929–2008. Thus, we arrive at the desired 2030s averaged climate-change modified flows $\overline{Q_m^{MF}} \approx \overline{q_m^M} + (\overline{Q_m^{NF}} - \overline{Q_m^{NH}})$ calculated in terms of available data.

Finally, in order to run the GENESYS adequacy model, a subset of water years of the observed modified flow $\{q_{mj}^M\}$ is chosen so that its weighted average, $\sum_j w_j [q_{mj}^M]$, could approximate $\overline{Q_m^{MF}}$. These optimal water-year weights $\{w_j\}$ are calculated by minimizing the objective function $OF$,

$$OF = \sum_{m=1}^{14} \left[ \left( \sum_{j=1929}^{2008} w_j q_{mj}^M \right) - \overline{Q_m^{MF}} \right]^2 \tag{1}$$

subjected to a normalization condition

$$\sum_{j=1929}^{2008} w_j = 1 \tag{2}$$

However, in practice $0.98 \leq \sum_j w_j \leq 1.02$. The optimized weights $\{w_j\}$ then determine the relative frequencies of water-year chosen in GENESYS simulations. The set of optimized weights $\{w_j\}$ then leads to an approximation for the average, $\sum_j w_j[q_{mj}^M] \cong \overline{Q_m^{MF}}$, with an error term whose square is just the objective function $OF$ in Eq. (1). If the error is small, then that subset of $\{q_{mj}^M\}$ could well represent the set of 2030's climate-change modified flows $\{Q_{mj}^{MF}\}$ to use in the GENESYS adequacy model, if their population distributions also have enough overlap. However since $\{Q_{mj}^{MF}\}$ is not available for direct comparison with subsets of $\{q_{mj}^M\}$, their non-modified flow equivalents $\{Q_{mj}^{NF}\}$ and $\{q_{mj}^N\}$ are compared in Supplementary Figures 3, 4 for five climate models.

One limitation from the approximations and assumptions used in this analysis is that using an optimal set of historical flows to approximate the 2030s average climate-change flow $\overline{Q^{MF}}$ will sometimes result in a small set of representative historical water years (see Supplementary Figures 3, 4). From those figures, it could be seen that only the GFDL-ESM2M and the INMCM4 climate models have a large enough set of historical water years and enough population overlap between the set of historical flows $\{q_{mj}^N\}$ and the 2030's climate-change flows $\{Q_{mj}^{NF}\}$. Furthermore, the same two climate models also have acceptably small error ($\sqrt{OF}$ in Eq. (1)) from using the optimized subset of $\{q_{mj}^M\}$ to approximate $\overline{Q^{MF}}$. For this reason, we decided to apply the resampling approach only for GFDL-ESM2M and INMCM4, which also happen to be the most conservative climate models in terms of changes relative to the mean historical monthly streamflow (see Supplementary Figure 2, the RCP8.5 and 2030s panel). Another limitation is the assumption that irrigation and evaporation amounts in future periods would be close to those in the historical period. Finally, the optimal set of historical flows was chosen to closely match the average climate-change flows only at the Dalles to represent the entire Columbia River Basin. However, analysis in the RMJOC report shows that climate-change impacts from some scenarios vary significantly throughout the Columbia River Basin.

**Performance measures.** GENESYS records all relevant data for each hour when generation fails to meet load. All adequacy metrics are computed using a subset of this data. Loss of load probability is the probability of incurring a shortfall event in any year. Since each simulation is of length one year, the Loss of Load Probability is computed as simply the percentage of simulations that incur any period of shortfall. The Loss of Load Hours (reported in Supplementary Table 1) is the expected number of hours per year when load exceeds generating capacity. This is simply the total number of hours of shortfall divided by the number of years simulated. The Expected Unserved Energy is a similar measure, computed as the total energy unserved during all shortfall periods, again divided by the number of years simulated (also reported in Supplementary Table 1).

Two additional metrics are added to complement the above-standard reporting metrics. These are the average maximum shortfall and the average event duration (AED). These differ from the Expected Unserved Load and the Loss of Load Hours in that they measure the nature of the average event incurred without being affected by the frequency of occurrence. The Average Maximum Shortfall is simply the average highest single-hour shortfall (in MW) across all simulated shortfall events. Similarly, the Average Event Duration is the average duration (in hours) across all simulated shortfall events.

**Code availability.** Power system simulations were run using GENESYS, a bespoke power system model maintained by the Northwest Power and Conservation Council. GENESYS is publically available by request to the Northwest Power and Conservation Council (https://www.nwcouncil.org/energy/energy-advisory-committees/system-analysis-advisory-committee/genesys-%E2%80%93-generation-evaluation-system-model). Post-processing R scripts for metric computation, bootstrapping, and jitter plots are available from the corresponding author on reasonable request.

## Data availability
Climate data used in this study were generated under the auspices of the River Management Joint Operating Committee (RMJOC) and are freely available for download at http://hydro.washington.edu/CRCC/data/. The datasets generated and analyzed during the current study are available from the corresponding author on reasonable request.

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

## Acknowledgements

We acknowledge the World Climate Research Programme's Working Group on Coupled Modelling, which is responsible for CMIP, and we thank National Center for Atmospheric Research (NCAR) for producing and making available their model output. For CMIP, the U.S. Department of Energy's Program for Climate Model Diagnosis and Intercomparison provided coordinating support and led development of software infrastructure in partnership with the Global Organization for Earth System Science Portals. This work was supported by the Office of Energy Policy and System Analysis of the U.S. Department of Energy. The Pacific Northwest National Laboratory is operated by Battelle for the U.S. Department of Energy under Contract DE-AC05-76RL01830.

## Author contributions

N.V. and J.F. designed the research. S.T., N.V., J.F., and D.H. performed the research. M.J. contributed the load forecast analytics. S.T. analyzed the data and led manuscript preparation. N.V. supervised the study. All named authors contributed to the writing of the manuscript.

## Additional information

**Competing interests:** The authors declare no competing interests.

