## [Peer Review File · Nature Communications]

Reviewers' comments:

Reviewer #1 (Remarks to the Author):

The authors perform an investigation of how the confluence of climate change effects on load (heating, cooling) and generation (particularly hydropower) in the U.S. Pacific Northwest affects power shortfall risk for the electricity system of the region.

The study is presented in a compelling way and provides interesting insights into the types of dynamics associated with simultaneous climate impacts on multiple factors and their translation to implications for electricity system planning, which is an important perspective to offer. The methods and statistical analysis are appropriate in detail for the scope of the study. The results and their explanations are easy to follow and clearly presented. With this given, I have a few minor questions / comments for the authors as follows:

1. The year 2035 is generally considered somewhat near-term as far as the temperature and precipitation severity of climate change impacts are concerned - meaning that the climate forcing expected to occur in that timeframe is often considered minor compared to longer-term studies focusing on 2050 and beyond. While this may not be true for all regions and models, for completeness, it may be useful for the authors to provide a statement on what the average changes in temperature and precipitation for the regions of interest by that timeframe to give the reader a sense of the scale of climate forcing that gives rise to the results.
2. Following on point 1, some discussion on how these results are expected to scale in future years may also be useful.
3. The selection of climate models is explained as the statistical methods mainly being valid for the most conservative models. Adding some discussion regarding the sensitivity of the results to the use of different climate models based on the general trends exhibited by them would also be useful. Additionally, more justification for the selection of the current climate models (other than method validity) will strengthen the paper.

Overall, the presented study provides a good contribution to the literature and an important scope. After addressing these minor comments, the paper will be suitable for publication.

Reviewer #2 (Remarks to the Author):

The paper discusses a potential seasonal reversal phenomenon in electrical power shortfall risk by quantifying several important performance measures in the future climate using GENESYS. The research problem is very important considering the ongoing process of climate change. This research method described in the paper is solid and the results are interesting and valuable. However, the major flaw of this paper is that it only concerns the modest climate change scenario in their research, but it is known that the climate change is with great uncertainty in the future. The paper should be with good quality once more RCPs are included, which not only aims to discover the uncertainty of the potential seasonal reversal in electric power shortfall, but also to test the robustness of the carbon risk resource expansion policy under the uncertainties of climate change. The paper can be considered fit to be published with major revision.

1. As described in Line 70, three conservative climate change projections are used for the six separate studies. What are the chosen "modest" climate change scenario here, RCP 4.5 or 6.0? The authors should give explanations. Then in Line 75, the water availability under climate change is determined using RCP 8.5 scenario, which I suppose is not the "modest" scenario used in the power system study. Why the water availability study is studied independently from the power system scenario? How the discrepancy of the use of climate change scenario in power system and water availability tells a consistent story?

2. As the introduction of the performance measures are given in later part of the paper, readers will not be able to understand the abbreviations describe in Table 1 (e.g. EUE, LOLH). These terms should be defined in the first place when they occur in the paper.

Reviewer #3 (Remarks to the Author):

This paper assesses electrical power shortfall for the Pacific Northwest region of the U.S. taking into account of both climate change impacts on loads and hydropower generation. Results are clearly articulated with supporting data but some of the claims need to be strengthened and discoursed further. This paper is well written, although a few clarifications up front would greatly help the reader.

Here are my main comments.

1. Are there other studies that also look at power shortfalls in this region or in the US? It would be very interesting to quantitatively, if not qualitatively, compare shortfall risks from this study with results from similar studies. I understand that this study investigates the combined supply and demand impacts of climate change, whilst previous studies may have focused on either supply or demand. I believe it is still worth the effort to see how this study differs from studies that examine only one side of the impacts.

2. My overall comment on Table 1 and other quantitative results is how robust are these numbers. Are the results under different scenarios significantly different? Some of the numbers have two decimal places, how reliable are they? What are the uncertainty bounds around these values?

3. You have only one climate change scenario (RCP8.5) to assess climate change impacts. Would you expect your results to differ substantially if you had picked RCP4.5 or another scenario?

4. On page 3, figure 1, the 3rd figure. Is it possible that there are missing x labels for the bars? I cannot figure out what the green, blue, and red bars represent from the text. In addition, a few sentences explaining this bar chart and the two policies in the main text would be really helpful.

5. On page 7, figure 2. What is the reason for no winter shortage events identified with IMNCM model?

Some minor comments.

Page 2, line 49. Delete "the".

Page 2, line 50. Add a citation for the claim of "collectively provide about half of overall annual generating capability".

Page 5, Table 1(a). This is the first time "EUE" appeared in the paper. Please define it up front. I understand it is defined in the supplementary, but new terms need to be defined when it first appears.

Page 18, reference 12 is incomplete.

Reviewers' comments:

We wish to thank the reviewers for their constructive feedbacks which have helped greatly improved the manuscript.

Reviewer #1 (Remarks to the Author):

The authors perform an investigation of how the confluence of climate change effects on load (heating, cooling) and generation (particularly hydropower) in the U.S. Pacific Northwest affects power shortfall risk for the electricity system of the region. The study is presented in a compelling way and provides interesting insights into the types of dynamics associated with simultaneous climate impacts on multiple factors and their translation to implications for electricity system planning, which is an important perspective to offer. The methods and statistical analysis are appropriate in detail for the scope of the study. The results and their explanations are easy to follow and clearly presented.

Thank you for the positive review and constructive feedback.

With this given, I have a few minor questions / comments for the authors as follows:

1. The year 2035 is generally considered somewhat near-term as far as the temperature and precipitation severity of climate change impacts are concerned - meaning that the climate forcing expected to occur in that timeframe is often considered minor compared to longer-term studies focusing on 2050 and beyond. While this may not be true for all regions and models, for completeness, it may be useful for the authors to provide a statement on what the average changes in temperature and precipitation for the regions of interest by that timeframe to give the reader a sense of the scale of climate forcing that gives rise to the results.

We have added two figures to the supplementary information showing (1) changes in annual maximum and minimum temperatures (it's the summer max and winter min that drives shortfall risk) and (2) changes to the flow regime (which is more closely related to the results than precipitation, given the contribution of snowmelt). These figures support some new statements in the paper relating to the unimportance of the RCP in the time slice of interest (lines 76 – 77), and the significant changes in climate forcing that one might expect to materialize as the century progresses (lines 213 – 215).

2. Following on point 1, some discussion on how these results are expected to scale in future years may also be useful.

We've included the maximum and minimum annual temperatures for the 2080s time slice in Figure S1, allowing readers to get an idea of the difference. The key point, which we've now added to the discussion (lines 213 – 215), is that the seasonal reversal that we observe under very modest temperature impacts would be expected to intensify in the future—particularly under RCP-8.5.

3. The selection of climate models is explained as the statistical methods mainly being valid for the most conservative models. Adding some discussion regarding the sensitivity of the results to the use of different climate models based on the general trends exhibited by them would also be

useful. Additionally, more justification for the selection of the current climate models (other than method validity) will strengthen the paper.

We have added some text to justify our model selection further (lines 74 – 75). The additional justification is that the use of conservative models adds to the robustness of the conclusions. Essentially, we're asking in this paper whether compound events may materialize under climate change. By demonstrating the presence of compound events under very modest change we are erring on the side of caution. It is reasonable to assume from newly added figures S1 and S2 that such effects would appear more intensely under the alternative scenarios.

Overall, the presented study provides a good contribution to the literature and an important scope. After addressing these minor comments, the paper will be suitable for publication.

Thank you again for your constructive comments and suggestions.

Reviewer #2 (Remarks to the Author):

The paper discusses a potential seasonal reversal phenomenon in electrical power shortfall risk by quantifying several important performance measures in the future climate using GENESYS. The research problem is very important considering the ongoing process of climate change. This research method described in the paper is solid and the results are interesting and valuable.

Thank you for your positive review and constructive suggestions.

However, the major flaw of this paper is that it only concerns the modest climate change scenario in their research, but it is known that the climate change is with great uncertainty in the future.

We agree that representing a large number of GCMs and RCPs is critical in climate impact studies focusing on understanding the range of responses. In our case however, we seek only to identify and demonstrate the presence of an interesting phenomenon—the possibility of heightened risk caused by compound events. Our conclusions are strengthened by the fact that this phenomenon appears so distinctly despite the fact that we consider only the most conservative changes in climate. When one looks at the patterns of temperature extremes and flows under more severe climate forcing (which we have now added as Supplementary Material), it is clear that the same pattern would occur—albeit with greater intensity—under those other scenarios. But adding these scenarios would not alter our conclusion whatsoever.

There is also a technical barrier to presenting results obtained by more extreme scenarios. Our power grid model relies on resampled historical inflows, which are connected to known hydropower operations for each year. To represent climate change impacts on hydropower dispatch, we resample historical annual flow profiles in such a way that reproduces average expected climatic change for each month. The problem is that when the changes to flow are dramatic, it's very difficult to represent those changes using resampled inflows. Typically we end up with only two or three very dry years being resampled, which means the flow inputs to GENESYS fail to capture the broad range of possible conditions represented by the GCMs (as noted in lines 357 – 358 of Methods). Only the most modest climate change scenarios could be

represented with confidence for our risk oriented analysis. Adding more GCMs would presently not lead to a robust representation of the range of responses. We have provided new figures in Supplementary Materials to support this point (Figures S3 and S4).

So, in summary, whilst we agree that a broad spectrum of climate models and RCPs is generally desirable in climate impact studies, we believe that this option would not further support our objective of demonstrating the concept of compounded events.

The paper should be with good quality once more RCPs are included, which not only aims to discover the uncertainty of the potential seasonal reversal in electric power shortfall, but also to test the robustness of the carbon risk resource expansion policy under the uncertainties of climate change.

Whilst we also agree that it is generally important to include a range of RCPs, we don't find that this will add significant value to the present study. The reason is that RCPs exert very little influence on the impact on temperature within the short time horizon used in our study (and in fact RCP8.5 turns out to be more conservative than RCP4.5 for the precipitation changes). We have added Supplementary Material to support this point (Figures S1, S2), which is also now noted in the method summary (lines 75 – 78).

Please note that the “carbon risk” policy is designed to be low cost in the event of local taxes on carbon emitting power generation. It is not necessarily designed to be robust under alternative climate futures. While testing the robustness of this policy under the uncertainties of climate change would be an interesting exercise, this is not the aim of the present work. Given the strict word limits of a Nature letter, we must focus on the study's main aims.

1. As described in Line 70, three conservative climate change projections are used for the six separate studies. What are the chosen “modest” climate change scenario here, RCP 4.5 or 6.0? The authors should give explanations. Then in Line 75, the water availability under climate change is determined using RCP 8.5 scenario, which I suppose is not the “modest” scenario used in the power system study. Why the water availability study is studied independently from the power system scenario? How the discrepancy of the use of climate change scenario in power system and water availability tells a consistent story?

This is miscommunication on our part—thank you for highlighting it. By “modest” climate change scenario, we are referring to the GCM rather than the RCP. We've rewritten this section to clarify (lines 70 - 92). Please note that the climate scenarios are fully consistent throughout this study.

2. As the introduction of the performance measures are given in later part of the paper, readers will not be able to understand the abbreviations describe in Table 1 (e.g. EUE, LOLH). These terms should be defined in the first place when they occur in the paper.

Agreed. We have added short descriptions of these metrics at an earlier point in the paper (lines 95 – 99), with the detailed description retained in the method section.

Reviewer #3 (Remarks to the Author):

This paper assesses electrical power shortfall for the Pacific Northwest region of the U.S. taking into account of both climate change impacts on loads and hydropower generation. Results are clearly articulated with supporting data but some of the claims need to be strengthened and discoursed further. This paper is well written, although a few clarifications up front would greatly help the reader.

Thank you for the positive and constructive feedback. We appreciate your suggestions for improving the clarity and impact of our study.

Here are my main comments.

1. Are there other studies that also look at power shortfalls in this region or in the US? It would be very interesting to quantitatively, if not qualitatively, compare shortfall risks from this study with results from similar studies. I understand that this study investigates the combined supply and demand impacts of climate change, whilst previous studies may have focused on either supply or demand. I believe it is still worth the effort to see how this study differs from studies that examine only one side of the impacts.

We agree that this type of comparison would be a valuable addition. However, whilst there are many academic studies that examine possible impacts of climate change on the performance of power generation from existing technologies (e.g., thermal, hydro), we find very few that address the resulting power system performance or reliability using a grid model. We think this is a major gap, because—as we state in the paper—the impacts on load or supply must be translated to shortfall risk if we are to understand associated risk to power grids. We've added a passage to the discussion to compare our findings with those of two similar studies performed for central Europe and eastern United States respectively (lines 201 – 210). But the key point that we make is that such studies are still rare.

2. My overall comment on Table 1 and other quantitative results is how robust are these numbers. Are the results under different scenarios significantly different? Some of the numbers have two decimal places, how reliable are they? What are the uncertainty bounds around these values?

These insightful questions have forced us to rethink how to present our results. We have decided to examine the robustness of the results (and the significance of the differences) using the bootstrap with 1000 repetitions. We resample (with replacement) the 6000 simulations for each study to produce uncertainty distributions for each of the reported metrics. We have decided to present results not as a table, but as box plots, allowing readers to visualize the robustness of the differences found. We believe this is now a much clearer and more robust presentation of our findings. We think the table is still useful, so we've moved it to Supplementary Material.

3. You have only one climate change scenario (RCP8.5) to assess climate change impacts. Would you expect your results to differ substantially if you had picked RCP4.5 or another scenario?

We don't believe that the RCP would make a substantial difference to our results, because the time horizon is relatively short. We've added some figures to Supplementary Material (Figures S1, S2) showing projections of peak summer temperatures (S1a), minimum winter temperatures (S1b) and streamflow (S2) for the Pacific Northwest, taken directly from a recent report from the River Management Joint Operating Committee (RMJOC). These include multiple GCMs and two

RCPs (RCP 8.5 and RCP 4.5). By 2035, the temperature differences between RCP scenarios have yet to really diverge, so we would not expect a substantially different result. We've now added this point to the main text (lines 75 – 78). The one interesting difference is in the flow under the GFDL model (surprisingly, the difference is greater under RCP4.5).

4. On page 3, figure 1, the 3rd figure. Is it possible that there are missing x labels for the bars? I cannot figure out what the green, blue, and red bars represent from the text. In addition, a few sentences explaining this bar chart and the two policies in the main text would be really helpful.

Thanks for highlighting this omission. The colors are consistent across all three panels, so they correspond to the labelling in the bottom-left panel. We now realize this isn't very clear, so we've added some additional labelling.

5. On page 7, figure 2. What is the reason for no winter shortage events identified with INMCM model?

For both the INMCM and GFDL projections, winter shortage events are reduced dramatically by warmer winter temperatures. However, the winter temperature impacts are similar for these GCMs, so at first glance it's not entirely obvious why INMCM results are slightly better off, with zero winter events recorded across 6000 simulations. The reason is that water availability is increased during winter under INMCM, and so hydro dispatch is enhanced. So this result may also be considered a combined impact (although it's less dramatic than the combined effects in summer of increased load and impaired hydro). We've added a sentence to highlight this point (lines 132 – 135).

Some minor comments.

Page 2, line 49. Delete "the".

Done.

Page 2, line 50. Add a citation for the claim of "collectively provide about half of overall annual generating capability".

Done.

Page 5, Table 1(a). This is the first time "EUE" appeared in the paper. Please define it up front. I understand it is defined in the supplementary, but new terms need to be defined when it first appears.

In the revised version of the paper, we've moved this table to the Supporting information. EUE is essentially a combination of LOLP and AMS, so it doesn't need to be included in the main body of the text. We keep it in the supporting information, however, because it is a standard supporting metric.

Page 18, reference 12 is incomplete.

Fixed.

REVIEWERS' COMMENTS:

Reviewer #1 (Remarks to the Author):

The authors have sufficiently addressed my comments on the first version of the manuscript. I recommend the paper for publication.

Reviewer #2 (Remarks to the Author):

The paper can be accepted for publication

Reviewer #3 (Remarks to the Author):

The authors have addressed all of my comments. I appreciate all the revisions.